# Does Trauma Change the Way Individuals with Post-Traumatic Stress Disorder (PTSD) Deal with Positive Stimuli?

**DOI:** 10.3390/bs14121195

**Published:** 2024-12-13

**Authors:** Olimpia Pino, Maurizio Rossi, Matteo Charles Malvezzi

**Affiliations:** Department of Medicine and Surgery, University of Parma, Via Volturno, 39, 43125 Parma, PR, Italy; maurizio.rossi@unipr.it (M.R.); matteocharles.malvezzi@unipr.it (M.C.M.)

**Keywords:** post-traumatic stress disorder, emotional regulation, valence system, reward system, trauma

## Abstract

Introduction: Post-Traumatic Stress Disorder (PTSD) is a highly prevalent disorder and a highly debilitating condition. Although current theories focused on depressed mood and intrusion as critical dimensions, the mechanism through which depression increases the risk of PTSD remains unclear. Research usually concentrates on the hyperactive negative valence system (NVS) (e.g., increased fear and threat responses), but some evidence suggests a significant role for the hypoactive positive valence system (PVS) (e.g., less neural activation towards rewards). Method: The main aim of the present research was to investigate whether probable PTSD leads to a different evaluation of the implicit processing in a refugee’s sample. Ratings of arousal, dominance, and valence from 60 International Affective Picture System (IAPS) pictures (positive, neutral, and negative) were collected from 42 individuals with probable PTSD, and a group of 26 trauma-exposed individuals (M_age_ = 28.49 years, SD = ±7.78). Results: ANOVA results revealed a main group effect (η2p = 0.379) on arousal, dominance, valence dimensions, and pictures’ categories (η2p = 0.620), confirming evidence according to which PTSD origins a state of maladaptive hyperarousal and troubles the regulation of emotions, and not supporting the view that such difficulties arise only with negative stimuli. Participants with probable PTSD deemed negative stimuli as more threatening than they really are, reacting to unpleasant images with greater negative emotionality (i.e., enhanced arousal and lower valence ratings) compared with individuals without PTSD. Moreover, they rated positive stimuli as less pleasant. Furthermore, arousal ratings were negatively correlated with valence (r = −0.709, *p* < 0.01) indicating that pictures with high arousal (negative) were associated with lower valence. Discussion: Our findings supported evidence according to which PTSD caused a constant state of hyperarousal and difficulties in regulating emotions facing environmental stimuli. Positive stimuli are considered less pleasant, and this inhibits from completely benefiting from them. Conclusion: Our study provides evidence for a differential and potentially complementary involvement of NVS and PVS in PTSD development. Intervention for PTSD may, thus, target both negative and positive valence processing.

## 1. Introduction

Post-Traumatic Stress Disorder (PTSD) was identified as a disorder in 1980 [1]. Being that 6–8% of the general population develops PTSD over the course of life, the diagnosis is frequent. Currently, this estimation has decreased compared to the past since the 5th edition Text Revision of the Diagnostic Statistical Manual (DSM) modified criterion A from the diagnostic criteria by excluding the traumatic event related to the disease and the traumas witnessed through the electronic media [2]. PTSD is widespread characterized by maladaptive fear responses following a traumatic event [3,4,5,6,7,8,9]. PTSD experienced by refugees has been linked to poor self-reported quality of life. Refugees are typically exposed to multiple types of traumatic events in their countries of origin. These events are often repeated, prolonged and interpersonal in nature, and have been demonstrated to have deleterious effects on mental health [9,10,11]. The effect of war on refugees’ lives is not temporary but persists over many years. Potentially traumatic events generally suffered by refugees and asylum-seekers in their countries of origin involve life-threatening injuries, witnessing death of loved ones, violence (interpersonal and sexual), and torture. The impact of exposure to traumatic events differs based on numerous issues such as area/country of origin, attributes of conflict, and personal factors including age, gender, ethnicity, or sexual orientation. The experience with potentially traumatic events contributes considerably to psychopathology [3,8,12]. Elevated rates of psychological disorders including Major Depressive Disorder (MDD) and PTSD are frequently reported among refugees and asylum seeker populations [13,14,15,16,17,18,19,20]. In addition, refugees face several daily challenges during displacement, as well as in the post-migration environment, including those related to lack of resources, family separation, social isolation and discrimination, socioeconomic factors, and immigration and refugee policies [8,14,16,21,22,23,24,25,26,27]. The growing number of displaced individuals worldwide and their disproportionately high rates of mental disorders have prompted the World Health Organization (WHO) to call upon treatment services to be responsive to the needs of asylum seekers and refugees [28,29].

Considerable evidence sustains abnormalities in affective processing. Psychophysiological research provided evidence for increased amygdala activation and decreased activation in prefrontal cortical areas as well as reduced hippocampal volumes in PTSD, leading to a hypothetically over-reactive threat detection, possibly diminished regulatory control, and disrupted adaptive memory processes [8,27,30,31]. Furthermore, the hyperactive negative valence system (NVS)—with increased physiological reactivity to aversive stimuli and reduced habituation of these reactions—has been involved in PTSD symptomatology. However, the heterogeneity of symptom profiles may be coupled with distinct aspects. It was shown that hyperarousal symptoms may be associated with increased neural responsiveness to negative stimuli and difficulties downregulating emotions, with increased attentional focus to possibly threatening stimuli constantly demonstrated [32,33]. Conversely, findings suggested that re-experiencing symptoms may be related with effortful suppression of intrusive emotions and thoughts. Moreover, avoidance and numbing may manifest as an overall disengagement from emotional processing with reduced neural responsiveness [34,35,36]. To date, popular theoretical models of PTSD and psychophysiological research have mainly focused on the inhibition system—fear learning, maintenance, and extinction—due to the nature of the disorder’s fear-related symptoms [15,27,28,29,30]. Even though less explored, PTSD also entails depressive symptoms, such as emotional numbing, which is associated with the reward system [15,27,28,29,30,35,36,37]. Evidence concerning the involvement of the reward system in PTSD is not definite regarding whether deficits in this system are entirely distinct from negatively valenced symptoms [3,31,32,34]. The reward system is a set of neural circuits that process appetitive stimuli—containing the limbic system (septal area, thalamus, hypothalamus, amygdala), basal ganglia (encompassing the ventral and dorsal striatum), ventromedial prefrontal cortex, ventral tegmental area (VTA), and substantia nigra. This system depends on neurotransmitters including serotonin, dopamine, and norepinephrine. Both opioid and cannabinoid systems are involved [3,38]. Some evidence suggests that PTSD might imply hypoactive positive valence system (PVS) (e.g., less neural activation towards rewards), as revealed by defective reward anticipation, diminished approach (reward-seeking) behavior, and reduced hedonic responses to rewards [27,34,35,36,37,38,39]. Abnormal reward-seeking and risky behaviors are in the PTSD criterion E2. Notably, PTSD is also highly comorbid with substance abuse (e.g., alcohol, nicotine, cannabis, and opioids). Elevated levels of external reward seeking may be an indication of reward system dysfunction in PTSD [40,41,42]. On the other hand, this maladaptive dimension in PTSD could be the consequence of an enhanced responsivity to negative stimuli [33,43,44]. In this perspective, considerable evidence relates this condition to oversensitivity of the negative system, which causes among PTSD people an amplified response to threatening stimuli [45,46] with symptoms of both hyperarousal and intrusion (i.e., re-experiencing) [7,9,34]. The negative valence system (NVS) involvement in PTSD onset has been consistently recognized as an abnormal activation of the salience network [7,8,9,10,11,12,13,14,15,16,17,18,19,20,21,22,23,24,25,26,27,28,29,30,31,32,33,34,35,36,37,38,39,40,41,42,43,44,45,46,47,48,49]. PTSD appears to be linked with biased neural valence processing, as indicated by hyper-responsivity to negative aversive stimuli and hypo-responsivity to positive rewarding stimuli [24,50,51,52,53,54,55]. However, depending on the involved methodologies and paradigms, the relative contribution of emotional processing to the development of PTSD remains mostly unknown [21,34,35,56,57,58,59,60,61,62,63] and there is still a need for further research to form a more definite picture [48,64,65,66,67,68]. Properly characterizing PTSD symptoms associated with the reward system could empower professionals to effectively recognize predictive factors for the disorder, appropriately diagnose it [39,42,50,62,68,69,70,71] and develop treatment programs that go beyond reversal of fear, sustaining the prevention of weak outcomes, such as risk behaviors [66,69,72,73,74]. On the other hand, some evidence does support the existence of reward learning deficits in the neural circuits of individuals with PTSD [39,45,56,57,58,59]. Accordingly, addressing these issues may guide research toward a better understanding of mental disorders and their underlying psychological, neural, and biological mechanisms, ultimately leading to improved treatments through diagnostic specificity, which is essential to develop precise interventions [14,18,28,29,53,69,75,76].

## 2. The Present Study

With the aim to explore implicit processing abnormalities of affective pictures in PTSD and potential underlying dimensions [24,35,39,61,63,65,67], our focus is on the population of refugees and asylum seekers. We were able to examine individuals having no history of diagnosis and treatment involving psychoactive medication or drug abuse. By comparing individuals having a probable PTSD with people who had experienced identical events but did not fulfill PTSD criteria, we intended to disentangle between effects related to trauma exposure and those related to the presence of PTSD. We explored the following questions: (1) What is the effect of probable PTSD on implicit affective evaluations of positive, negative, and neutral items? (2) What is the effect of trauma exposure on affective ratings? With respect to the presented literature [36,39,42,50,51,53,54,57,60,67,70], we suggested that if the disorder involves a numbing, then participants with probable PTSD would also show lower valence ratings (i.e., less pleasure affect) and/or larger ratings of arousal/activation (i.e., reduced emotional activation) in response to negative photographs compared with the trauma-exposed individuals.

### 2.1. Method

#### 2.1.1. Research Type and Design

We employed a cross-sectional study to investigate the emotional differences in valence (unpleasant/pleasant), arousal (calm/agitation), and dominance (domination/non-domination) in individuals diagnosed with PTSD from those who are not. We focus on immigrants and refugees, as this population has been frequently exposed to severe potentially traumatic experiences. The design of the study was a 2 (Participants: Probable PTSD Group, Trauma-exposed Group) × 3 (Item type: Positive, Neutral, Negative) × 2 (Affective ratings: Arousal, Dominance, Valence) design, with all factors being between-participants.

#### 2.1.2. Population, Sample and Sampling

For the current study, a convenience sample of 69 trauma-exposed people was recruited between September 2019 and February 2020 at the Coop Dimora D’Abramo refugee’s center from Consorzio di Solidarietà Sociale Oscar Romero established in Reggio Emilia, Italy. Inclusion criteria were defined as the following: (a) age between 18 and 60 years; (b) status as a refugee or asylum seeker, which is defined according to the UNHCR (UNHCR, 2018); (c) exposure to relevant potentially traumatic events; (d) informed consent. The exclusion criteria were as follows: (a) missing informed consent; (b) age under 18 years or over 60 years; (c) current risk of suicidality established on clinical judgment; (d) self-reported as having a color-vision deficit. Following enrollment, participants were further subdivided according to PTSD symptoms (Probable PTSD Group vs. Trauma-exposed Group).

### 2.2. Materials

#### Selection of the Experimental Stimuli

Pictures employed in the present study were selected from the International Affective Picture System (IAPS) based on normative ratings of arousal, dominance and valence, which is currently used in several research fields. The IAPS is a catalog of photographic images that have been shown to induce positive, negative, or neutral affective states [77,78]. Converging evidence that discrete basic emotions have consistent and discriminable neural correlates was also provided [33,35,54,67,79]. The purpose of the IAPS is to propose standardized emotional visual stimuli. The construction of the IAPS was based on Lang and Bradley’s model of the two dimensions of emotions: valence and arousal [78]. Motivational aspects of behavior, such as approach to appetitive stimuli and avoidance of aversive stimuli, can be described as computable parameters of the pleasantness/unpleasantness (valence) and intensity of emotional activation (arousal), which express a hypothetical two-dimensional affective space [77]. The process of validating the pictures from the IAPS involves applying a self-report method, the Self-Assessment Manikin (SAM) [79] that delivers knowledge on. It presents three-dimensional values of arousal (the level of excitation), dominance, and valence (pleasantness or unpleasantness) of emotional responses, which are assessed through a standardized assessment in terms of valence (ranging from “pleasant” to “unpleasant”), arousal (ranging from “calm” to “excited”), and dominance (ranging from “in control of emotion” to “influenced by the emotion evoked by the picture”) with nine points for each scale. Plotting mean valence and arousal ratings of each picture on a Cartesian plane, they are organized as a “boomerang” pattern where the upper arm designates appetitive motivation (“approach-like”), and the lower one shows aversive motivation (“avoidance-like”). The normative ratings of the images were reported as a function of the average of the three dimensions per picture obtained in the original study and validations carried out in several countries [77,80].

Sixty photographs were chosen for the present study based on their normative rating from the IAPS [77]. Among the selected items, 28 presented positive events, 20 pictures with negative events, and 12 displayed neutral events. They had different average values: 28 of positive valence (M = 7.008, SD = 0.796), 20 of negative valence (M = 3.001, SD = 0.981), and 12 of neutral valence (M = 4.991, SD = 0.312). The arousal levels were on average 4.693 (SD = 0.959) for the positive images, 5.799 (SD = 0.948) for the negative ones, 4.604 (SD = 1.288) for the neutral ones. The dominance levels were on average 5.672 (SD = 0.717) for the positive images, 3.743 (SD = 0.699) for the negative ones, and 4.971 (SD = 0.604) for the neutral ones.

### 2.3. Instruments

#### 2.3.1. Questionnaire Measuring Demographics and Trauma Exposure

Exposure to potentially traumatizing events was determined by a semi-structured interview. Items involved traumas directed at the self and others. This interview collected information about socio-demographic (participants’ gender, age, education, marital status, religion, years in Italy, living with family, employment), and trauma- and displacement-related factors. The list of potentially traumatic experiences was based on events frequently experienced by refugee and asylum seekers [72,81,82] and aimed to collect as much information as possible (in total, 13 categories of traumatic events were defined). These events contained torture, assault, political imprisonment, witnessing murder, witnessing to genocides, dead/separations of family or close others, and severe deprivation of medical care for self or others. We also entered an inventory of adversities such as insufficient food, inadequate finances, poor shelter, unemployment, and experiences of conflict. Participants indicated experiencing and/or witnessing each type of event through “yes/no” responses.

#### 2.3.2. PTSD Symptom Screening

To detect PTSD symptoms, the revised Part IV from the Harvard Trauma Questionnaire (HTQ) that contains 40 items related to PTSD and refugee-specific expressions of functional distress and trauma was used. The HTQ, originally developed by Mollica and colleagues [81], demonstrated good psychometric properties, with an inter-rater reliability of 0.93, a test–retest reliability of 0.89 (*p* < 0.0001), and internal consistency reliability of 0.90. Furthermore, other studies evidenced a robust internal consistency for the fourth part (0.96). Similarly, the criterion validity analysis suggested a threshold of 2.5 with sensitivity of 0.78 and specificity of 0.65 [81,82]. However, the widespread modification of the diagnostic criteria for PTSD has necessitated the modification of Part 4 according to the DSM-5 criterion for PTSD (HTQ-5). Vindbjerg and colleagues [83] have proposed a first revision of the original tool (HTQ-PTSD-R) in which, basing on the DSM-5 criterion, the 16 items were allocated into the following four clusters: (a) Intrusion symptoms; (b) Avoidance; (c) Negative alterations in cognitions and mood; (d) Alterations in arousal and reactivity or into two subscales, an arousal-intrusion subscale (AIS) and an avoidance-numbing subscale (ANS). The main updates of the current version [84,85] were the addition of nine new items including two dissociative specifiers (items 17–25). Confirmatory factor analysis (CFA) of the one-factor solution revealed acceptable goodness-of-fit statistics for RMSEA (RMSEA = 0.077, CFI = 0.72, TLI = 0.68), also providing evidence for one-dimensionality, and acceptable internal consistency with Cronbach’s alpha of 0.70 [86]. The HTQ has been found to be valid and reliable across a large range of populations [81,82,85,87,88,89].

The HTQ-5 was employed for the purposes of the present investigation. Participants are requested to report on a four-point Likert scale how much they had been troubled by a distinct symptom, ranging from “not at all” (1) to “extremely” (4), during the last week. The occurrence and severity of PTSD symptoms were explored using the mean scores of the HTQ-25 items. A standard cut-off score of 2.5, as indicated by Mollica et al. [81] was considered to suggest probable PTSD for participants who responded in English. The Italian validation of the tool by Pino and colleagues [90] has showed an excellent internal consistency, revealing a HTQ five-factor solution as the best model, with satisfactory indexes of fit and, basing on sensibility (0.963) and specificity (0.189), the best cut-off of 2.0 allowed discriminating for PTSD positive cases. This version was used for participants who preferred conducting the study in Italian language. Individuals can be regarded as symptomatic if their mean score achieves the cut-off of ≥2.

### 2.4. Affective Ratings

Figure 1 shows the experimental procedure. To ensure both reliability and validity of the affective ratings when applied on a new sample, researchers are requested to follow instructions from the normative rating procedure manual handled in the original investigation and extended studies [77,78,79,91]. To evaluate the three dimensions of pleasure, arousal, and dominance, the Self-Assessment Manikin (SAM) [78,79] was employed. It consists of a graphic representation showing values along the three dimensions on a continuously varying scale that depicts emotional reactions. According to Bradley and collaborators [79], the valuation of each dimension arranges the response of the individuals to affective stimuli, which can be theorized as two essentially motivational systems of avoidance (defensive system) and approach (appetitive system). The defensive system is mainly stimulated in situations signifying a threat to survival and safety, producing reactions such as withdrawal/attack/freezing, whereas the appetitive system is predominantly activated in conditions that produce pleasure and well-being. These two systems account for two basic dimensions of emotion: valence and arousal. The first specifies which system is working, while the second reflects the intensity of the activation. Hence, the emotional reaction prompted by affective stimuli may be described by its position on a two-dimensional affective space (ordinate = valence; abscissa = arousal). This way, the subjects exposed to the IAPS photographs can select any one of the images included on each scale or from between either of the two images that results in a scale of nine points for each dimension [78]. Ratings are scored such that 9 represents a high evaluation on each dimension and 1 represents a low rating on each dimension. The dimension of valence was employed to evaluate the degree of positive or negative reaction induced by a given picture, ranging from 1 to 9 (1 for very negative emotions, 9 for very positive emotions). Regarding the arousal scale, participants are required to assess the extent to which a given image makes them feel unaroused or aroused, ranging from 1 to 9 (1 for unaroused/relaxed and 9 for very much aroused, that is agitated or excited). Finally, the dominance dimension spans from “out of control” to “in control” and is, respectively, denoted by a tiny shape or as an enormous shape.

Another study confirmed valence ratings to be like the previous norms while participants were more prone to rate the pictures as less arousing than in the former norms. The mean ratings on the remaining dimensions were all below the midpoint of the 9-point Likert scale. A variability in ratings across the pictures suggested that selecting slides based on these variables is possible. The means and standard deviations of emotional ratings may be downloaded from electronic supplementary material in [92].

### 2.5. Procedure

The participants were informed about the aim of the research, and no identification or names were recorded to maintain confidentiality. The subjects’ consent was obtained after explaining information on the experiment. The study participants were informed of their right to refuse or stop participating at any time. The experimental sessions were completed by psychologists with master’s or doctoral-level qualifications, with the aid of experienced interpreters, when necessary. In the first session, the purpose of the study was explained to all participants and written consent was obtained. Once the information letter was read and the consent form signed, the semi-structured interview and the HTQ were administered in English or in Italian with the support of cultural mediators. Following this and the allocation to groups, a second session took place. Participants were seated in front of a white screen and were informed that the aim of the study was to investigate how humans respond to pictures that represent different events occurring in life. Following this, the instructor stated the detailed explanation of the procedure, the instructions were presented, and the photographs were then shown sequentially. The experimenter first explained the meaning of valence, arousal, and dominance to the participants and then how to rate the pictures on a paper version of the SAM. Instructions were provided to specifically target the affective response associated with each stimulus. Pictures were presented on a monitor. Participants were instructed to rate how they felt while looking at each picture. A total of sixty pictures was presented with the following sequence: a warning slide displayed for 5 s, informing of the number of the picture to be rated. The picture itself followed and was shown for 6 s. Participants then had 15 s to rate the dimensions of valence, arousal, and dominance for each picture using a 9-point SAM [79].

The valence section measures the participant’s emotions along a range from pleasant to unpleasant, the arousal segment evaluates emotions from calm to excited, and the dominance explores emotions going from controlled to uncontrolled states. Before starting the procedure, subjects went through two practice trials with the aim of familiarizing themselves with the rating procedure: one positive and one negative. This provided the subject with an overview of the emotional content of the pictures that would be administered during the study and functioned as an anchor for the affective ratings. Before viewing a given IAPS image, the instruction: “Please, rate the next image in line number n…” appeared in the screen for 5 s. During this time, the participant must find in her/his booklet the numerical code relating to the row where that image should be evaluated. After 5 s, the first IAPS photograph was displayed for 6 s. During this interval, the subject was instructed to give attention to the picture and the suggestion “Please rate the picture in the three dimensions now” appeared into the screen for 15 s. Throughout this time, the participant rated the stimulus according to the three dimensions in her/his booklet. When the interval was passed, the ensuing trial initiated. The pictures were administered in a random arrangement following limitations: no more than two pictures from each affective category (negative, positive, neutral) and no more than three images from each type occurred sequentially. The IAPS pictures were displayed on the whole screen (see [77]). These procedures were repeated until all 60 IAPS pictures were rated. Each trial took 26 s and the whole process took roughly 30 min. Participants who were found to have a probable PTSD received information on their condition, and they were forwarded to a nearby psychiatric unit.

### 2.6. Data Analysis

This study aimed to explore the differences in affective responses among trauma survivors, categorized as either having “Probable PTSD” or being merely “Trauma-exposed”. The PTSD classification was determined using the HTQ-5, with cut-off scores set at 2.5 for the English version and 2.0 for the Italian version (refer to Measures section). Emotional normality distribution was assessed with Kolmogorov–Smirnov test. Categorical data were represented with counts (n) and percentages (%), while continuous variables were reported as mean ± standard error (ES). For categorical variables, Pearson’s chi-square and Fisher’s exact tests were employed when the chi-square assumptions were not met. Continuous variables were compared using Student’s *t*-test, provided assumptions of normality and equal variances held; otherwise, the Mann–Whitney U test was used.

Firstly, for each image, the mean valence, arousal, and dominance ratings and the corresponding standard deviations (SD) were calculated. With the purpose of exploring the influence of PTSD and picture category on the three emotional dimensions (Valence, Arousal, and Dominance) a series of 2 (Group: Probable PTSD vs. Trauma-exposed) × 3 (Type of images: positive vs. neutral vs. negative) ANOVAs were separately carried out on each emotional rating. The effect sizes were estimated using partial eta-squared (η2p). Trauma occurrence/exposure and type of images were treated as between-group factors. Lastly, Spearman’s correlation coefficients (r) were calculated to examine the relationships among valence, arousal, and dominance ratings within the sample. All analyses were performed using statistical software, IBM SPSS Statistics for Windows, Version 29.0. Armonk, NY: IBM Corp. The statistical threshold was set at *p* < 0.05.

### 2.7. Ethics

Study procedures were approved by the local Ethic Committee of the Area Vasta Emilia Nord—AVEN (Ref. 1156/2020/OSS/UNIPR) and were carried out in accordance with the Declaration of Helsinki. After receiving a description of the study’s aims and procedures and being given the chance to ask questions, written informed consent was obtained from all individual participants included in the study.

## 3. Results

### 3.1. Sample Characteristics and PTSD Occurrence

Detailed information concerning the main sociodemographic and clinical characteristics assessed is provided in Table 1 where mean and percentage of the main variables were reported. The present study included 68 participants; one subject was excluded due to incomplete data. Their mean age (±SD) was 28.493 (±7.785) years, with age ranging from 18 to 53 years. Among the respondents, 58 (85.15%) were male and 52 (78.45%) were single. Most of the refugees in the sample named Nigeria as their country of origin (25%), followed by Pakistan (14.71%), The Gambia (11.8%), Bangladesh (8.82%), and Ivory Coast (8.82%). In total, 44% percent of participants completed middle school followed by 29.4% without any qualification. Most of them are Muslim by religion (63.2%) followed by 35.3% of Christian religion. The most frequent means of arrival in Italy was by boat, followed by on foot (4%) and by train (4%), and before Italy, 89.7% had been in other countries. The majority were asylum seekers (86.4%) whereas 13.55% were of refugee status. The mean years of stay in Italy was of 3.74 years (SD = 2.00). Regarding the current occupation, 44 participants (64.7%) were currently not occupied while 53 of them (77.9%) worked in their country of origin. All participants fulfilled the DSM-5 criterion “A” having been exposed (i.e., experienced, or directly witnessed) to several types of traumatic events. The most frequent type of trauma experienced included threats (72.1%), having witnessed beatings of acquaintances (41.2%), unknown people (33.8%) or acquaintances’ rape (20.6%). Having witnessed death of acquaintances (54.4%), injured people (36%), suffered injuries themselves (32.4%), especially attributable to beatings (33.8%) and killing of relatives (25%) were also common. On average, trauma exposure occurred 5.44 years (SD = 3.61) earlier.

Of the 68 participants, according to HTQ-5 scores, 26 met criteria for probable PTSD (Probable PTSD group, 22 Males) whereas 42 did not meet criteria for PTSD even though they had experienced significant traumatic experiences (Trauma-exposed Group, 36 Males). The sociodemographic characteristics appear to be highly comparable between groups. The groups did not differ in age, education level, and other characteristics of trauma exposure but only for scores from Part IV of the HTQ-5 for PTSD [t (66) = 11.66, *p* < 0.0001].

### 3.2. Effect of PTSD on Emotional Ratings and Type of Stimuli

Mean of emotional ratings for pictures as a function of groups (Probable PTSD, Trauma-exposed) and conditions (positive, negative, and neutral images) are reported in Figure 2. To answer research questions about the effect of PTSD on emotional ratings and type of pictures, SAM arousal, valence and dominance data were then analyzed in separate analyses of variance (ANOVAs), with group (Probable PTSD, Trauma-exposed) and pictures category (positive, neutral, negative) as between-subjects factors. These effects are reported in Table 2.

With respect to the arousal ratings, those of Probable PTSD Group exceeded on average those of Trauma-exposed Group. They decreased across the picture’s category from negative (see Figure 2 panel a), to neutral, and positive. In the ANOVA, (see Table 2) a significant main effect of group (*p* < 0.001, η2p = 0.379) for the arousal dimension was found, with pairwise comparisons showing higher ratings for the Probable PTSD Group compared to the Trauma-exposed Group (*p* < 0.001). Basically, the important differences were that our participants with probable PTSD were more likely to rate the images as more arousing rather than participants of the Trauma-exposed Group. As initially expected, the picture category’s main effect was also significant (*p* < 0.001, η2p = 0.620) with pairwise comparisons revealing that negative pictures being rated as more arousing compared to neutral (*p* < 0.001) and positive (*p* < 0.001) pictures. Neutral images were additionally assessed as more arousing than positive images (*p* < 0.001), and the interaction is not significant.

In panel b of Figure 2, the mean dominance ratings for the Probable PTSD and Trauma-exposed Groups were indicated, respectively, for the different types of images (positive, neutral, and negative). The mean ratings of dominance for positive images are significantly lower for Probable PTSD participants than those produced by the Trauma-exposed subjects. Similarly, for neutral pictures, the Probable PTSD Group’s mean dominance ratings were poorer than those of the Trauma-exposed Group. A two-way ANOVA model of Group (Probable PTSD vs. Trauma-exposed) by Picture type (Positive vs. Neutral vs. Negative) on mean dominance ratings yielded a main effect (*p* < 0.001, η2p = 0.547) only for pictures’ category. Pairwise comparisons showed enhanced dominance ratings for positive pictures compared to neutral (*p* < 0.001) and aversive pictures (*p* < 0.001). Furthermore, neutral pictures displayed more dominance compared to negative ones (*p* < 0.001). The interaction between groups and picture types did not reach significance.

In panel c of Figure 1, the mean valence ratings for the Probable PTSD and Trauma-exposed Groups were shown, respectively, for positive, neutral and negative pictures. Individuals with probable PTSD responded to all images with lower evaluations relative to the controls for negative, positive, and for neutral pictures. Overall mean valence ratings were lower for Probable PTSD participants than Trauma-exposed participants. A two-way ANOVA model of Group (Probable PTSD vs. Trauma-exposed) by Picture type (positive vs. neutral vs. negative) yielded a significant main effect of group (*p* < 0.001, η2p = 0.315) suggesting that Probable PTSD participants rated valence significantly less than Trauma-exposed individuals and almost below the mid-point of the scale. The analysis yielded an additional main effect of picture category (*p* < 0.001, η2p = 0.788). Overall, positive images were valued as more pleasant than neutral (*p* < 0.001) and unpleasant (*p* < 0.001) ones. Furthermore, neutral images were also regarded as more pleasant than negative pictures (*p* < 0.001). The remaining interaction between groups and picture types was significant (*p* = 0.019, η2p = 0.58) indicating that valence ratings vary between groups as the types of images vary.

### 3.3. Analysis of Correlation Coefficients

To assess relationships within each group, between emotional ratings for each of the 60 images across all stimuli conditions, Spearman’s correlation coefficients (r) were calculated. The resulting findings can be seen in Table 3. To explore the exact contributions of valence and arousal in affective processing, the occurrence of negative correlations between these dimensions was also tested within groups, first for negative images, then for positive images and, finally, for neutral images allowing for the examination of group differences in appetitive and defensive motivation when viewing the IAPS.

Regarding the Probable PTSD Group, results showed that for negative pictures, the arousal ratings were negatively correlated with valence ratings (r = −0.709, *p* < 0.01), indicating that pictures with high arousal (negative) were associated with lower valence. Furthermore, Probable PTSD participants showed a significantly negative correlation for unpleasant pictures between arousal and dominance (r = −495, *p* < 0.05), suggesting that as the arousal value increases, dominance decreases.

Among Trauma-exposed emotional ratings, a negative significant correlation is yielded for unpleasant images between arousal and dominance (r = −515, *p* < 0.01), indicating, as happens for Probable PTSD participants, that negative aroused pictures were poorly dominant, and a significant positive correlation between valence and dominance was shown for negative pictures (r = 434, *p* < 0.01) suggesting that Trauma-exposed participants feel themselves dominated by the unpleasant pictures. The negative correlation between arousal and valence, although significant as for the other group, was weak (r = −368, *p* < 0.05).

## 4. Discussion

Our research aimed at investigating how the affective response differs between trauma survivors either suffering from probable PTSD or not. The main purpose of the present study was exploring whether a probable PTSD leads to a different rating of the emotional dimensions of valence (unpleasantness/pleasantness), arousal (calm/agitated), and dominance (domination/non-domination). We anticipated significant interactions across all trauma and emotional content conditions, except for neutral images. Specifically, it was expected that the Probable PTSD Group would exhibit the highest arousal levels in response to negative images compared to the control group [34,35,39,58,61,67]. However, given evidence on trauma’s impact on neural responses to reward systems, it was hypothesized that positive images would induce lower self-reported valence levels in the Probable PTSD Group. Refugees and asylum seekers arrive in our country after dangerous journeys often crossing through Libya, where they are subjected to intimidation, torture, or kidnapping before crossing the Mediterranean on precarious vessels. Furthermore, the participants often claimed to have witnessed violence of various kinds even against family and friends and/or strangers, events considered potentially traumatic events, as suggested by the criterion “A” from the DSM-5-TR. Many participants witnessed death firsthand, assisting family members, friends, or strangers at the time of passing, due to natural causes such as illness and old age, but also to wasting due to lack of food and water or extreme heat and tiredness during the journey. Finally, witnessing the murder of one’s travel companions was also common [8,10,11,12,13,16,18,20,25,27,55,72,73,74]. These especially harmful events probably favored the onset of PTSD. Indeed, the administration of the HTQ-5 DSM-5 PTSD [81,83,84,85,86,90] in our sample revealed the disorder among 38.2% of cases. This prevalence is in line with the literature according to which PTSD in refugees and asylum seekers shows a prevalence rate between 23% and 88.3% [68,75,82,83,88], much higher than in the general population where it is between 0.2% and 3.8%.

The results of exposure to visual stimuli of different emotional content (positive, negative, and neutral) have brought evidence supporting the anticipated hypotheses. Ratings differed as a function of group, and type of stimuli. Specifically, the Probable PTSD Group tended to have higher ratings in arousal dimensions than the control group. In particular, the statistical analyses carried out according to the pictures’ emotional dimension of valence (unpleasantness/pleasantness), arousal (calm/agitated), and dominance (domination/non-domination), showed how participants with the probable PTSD perceived themselves more agitated and with less control in front of any type of image than individuals from the Trauma-exposed Group. Conversely, Trauma-exposed participants had the highest dominance and valence ratings irrespective of picture’s type. For dominance ratings, higher reports of feeling “in-control” in the order of highest to lowest were reported across positive, neutral, and negative pictures. Also of note, group differences disappeared for dominance ratings with negative images. Valence scores decrease more for positive and neutral images, and unpleasant judgments are emphasized more for negative images.

The most striking finding concerning group differences was that Probable PTSD and Trauma-exposed participants showed a different pattern of ratings, which demonstrated that the Probable PTSD Group reported an overall higher arousal for all picture categories but lower dominance and valence than the Trauma-exposed Group when viewing IAPS images. These results suggest that when participants viewed IAPS images, they mostly reported similar affective responses in all categories but Probable PTSD participants’ responses when rating arousal were significantly different from those of Trauma-exposed Group. However, these effects were probably present for some IAPS pictures (e.g., human attacks, combat related injuries), but not all (e.g., family, nature); because the IAPS contains different images, this finding is not surprising for arousal dimension.

In the current investigation, group and picture type effects were also explored using distributions of affective space. The dimensions of valence, dominance, and arousal were also significantly correlated with each other. The level of valence was significantly associated with the arousal level among both participants groups for the negative pictures. For negative pictures, for both groups of participants the levels of valence were inversely associated with arousal and dominance. However, strong couplings between arousal and dominance were observed for Probable PTSD participants’ ratings while viewing neutral images. Furthermore, Trauma-exposed individuals tend to report with weak couplings between valence with arousal, and valence with dominance for negative pictures. Finally, a significant positive correlation was evidenced for the pleasant picture between valence and dominance. It can be supposed that for the positive pictures, the Trauma-exposed subjects were more relaxed the more pleasant the image was, whereas for the negative images, both groups of participants were more agitated and dominated the more unpleasant the image was.

The results confirmed evidence according to which PTSD causes a constant state of maladaptive hyperarousal and difficulties in regulating emotions facing environmental stimuli, regardless of whether they are positive, negative, or neutral, and disagree with those who argue that such difficulties arise only with negative stimuli [30,35,54,61]. It can be assumed that following exposure to several traumas in refugees/asylum seekers, the PTSD onset establishes a generalization of the troubles in emotions regulation even towards stimuli that do not directly recall the traumatic events [8,19,24,35,39,54,58,65]. Therefore, individuals with probable PTSD manage all stimuli as more threatening than they really are and feel themselves in an exaggerated state of agitation that does not allow them to give due weight to what they see [33,35,43,52,57,64]. Specifically, the negative valence turned out to be as a strong reminder that might have triggered memories related to the individual traumatic experience and induced flashbacks. Moreover, positive stimuli are considered less pleasant, and this prevents them from fully enjoying them [36,41,67,70]. Therefore, people who meet PTSD criteria exhibit great difficulties in regulating emotions [32,33,37,55,58,59]. This conclusion appears consistent with a six-factor model [68,71] according to which the symptoms’ cluster named “negative alterations in mood and cognitions” from DSM-5 was furthermore broken down into a first cluster concerning the negative affect (negative affect potentiation) and a second cluster about anhedonia (positive affect worsening). Noteworthy in this regard is that the Research Domain Criteria (RDoC), launched by the National Institutes of Mental Health (NIMH), suggested distinguishing positive valence from negative valence in the context of psychopathology [63,64,65,66,67,68,69,70,71].

## 5. Limitation and Future Directions

The research has several strengths. It included important variables that were not considered in previous studies, such as comparing individuals who developed a probable PTSD exposed to the same potentially traumatic events of individuals who did not develop traumatic symptoms. Hence, group differences can be exclusively ascribed to PTSD status since there were not considerable differences between participants’ groups. Furthermore, an updated standardized instrument for measuring PTSD according to DSM-5 was used [83,84,85,86,90]. Nevertheless, several limitations of this work must be acknowledged. All the participants were recruited in the same center in Italy and the sample size is relatively small. Moreover, excluding individuals hosted by relatives or friends or living in unregistered camps, could be considered as a form of limitation bias of the present research. Considering the magnitude of our results, statistical power, and the cross-sectional design, any generalization of the findings must be approached with caution. Furthermore, the probable presence of depression, whose effects on emotion regulation, were not verified and could be confused with those of PTSD. It could be useful to include, among the tools, a test to verify the possible presence of depression so as not to have possible interference effects between the two disorders, often comorbid with each other. The level of education held by the participants may have made the interpretation of the 9-point SAM scales difficult. Since the number of points did not correspond to the number of stick figures of the SAM, some participants found it difficult to assign scores, tending to give extreme answers on one side or the other of the continuum. The use of the 9-point SAM scale could be changed to the 7-point or 5-point version, to facilitate the understanding of the assessment task by the participants [79,92]. Some of the IAPS images selected in the study were challenging for the participants to interpret, especially those of neutral valence (e.g., mineral images) which may have been incorrectly interpreted. Evaluating emotional processing through subjective measures may be especially problematic for refugees and asylum-seekers because of their strain in recognizing and labelling their own emotions. Moreover, the setting itself may have functioned as a trauma reminder for several people, although we put great effort into ensuring that the research situation was not evocative of trauma experiences (e.g., only psychologists conducted the experiment).

In the future, the study could be repeated with a larger sample size with subjects from different centers. Moreover, neurobiological measures of emotion including physiological and neural measures should expand our understanding of how individuals react to emotional stimuli and regulate negative emotions to better describe clinical affective disorders [31,49,68,80]. Finally, it would also be interesting to be able to have a gender-balanced sample to verify whether women with asylum seeker or refugee status are predisposed to developing PTSD, as is typical of the general population. We anticipated that women would report higher levels of psychological distress than men. Men and women experience different migration circumstances as adverse experiences of arrival and settlement, with significant variation by ethno-national subgroup [34,59,65,73,87,93,94]. However, given the rather small sample size obtained, it was not possible to observe this differential effect of gender.

Nevertheless, it is meaningful to explore the relationship between PTSD in the context of processing of aversive and rewarding stimuli. The findings from the present investigation have potential clinical implications. Firstly, our study confirmed the possibility that standardized instruments accurately distinguish individuals with probable PTSD who need a more detailed diagnosis and intervention. People who are traumatized are much more likely to avoid looking at something upsetting than people who are not. The current diagnostic process is not as effective as it needs to be, and mental health tools are not adaptable to various identities, leaving many patients struggling for long periods. Furthermore, due to the nature of the disorder and the risk of re-traumatization, it is useful to employ rapid tools such as the HTQ and implicit ones such as the IAPS. The conclusion that a low emotional regulation (indexed by high arousal levels) in refugees and asylum seekers is related to a conceivable weakened anger recovery resulting from the administration of negative images can highlight the need to consider prolonged anger responses within the psychological intervention [4,6,8,12,14,15,23,39,45,50,55,56]. Furthermore, the present investigation emphasized that affective reactions in PTSD do not exclusively concern the evocation of negative effects but are also reflected on positive stimuli. Therefore, when focusing on an individual’s emotions, the precise evaluation of both is vital. This also applies to research addressing affective reactions following trauma cues, with these findings suggesting that it is worth assessing both positive and negative responses. Additionally, a relevant issue concerns the ability for the IAPS to evaluate emotional processing. Scholars have questioned the sensitivity of the IAPS ranking for the estimation of PTSD-related numbing, and recommendations have been proposed to increase the sensitivity of the tool by expanding the images amount and adding trauma-related photographs [35,80]. The finding that the IAPS shows differences between groups points to this pictures catalog can elicit abnormal affective responses in trauma-related conditions such as PTSD. Finally, instead of using pictures, as in this study, more complex stimuli, such as videos or virtual reality tasks, to investigate attention allocation could cause intense or different responses. Future investigations could use new technologies to estimate attention patterns in more complex threatening situations. In addition, these procedures can be adapted to explore PTSD populations using functional neuroimaging procedures to further examine the physiological mechanisms of emotion-related disorders. Despite the above limits, these findings add to the knowledge within the trauma field, which involves the role of impaired emotional processing in people with PTSD.

## 6. Conclusions

This study described the level of PTSD and its associated factors, among refugees and asylum seekers residing in Italy exploring the emotional differences in valence (unpleasant/pleasant), arousal (calm/agitation) and dominance (domination/non-domination) in individuals diagnosed with PTSD from those who are not. There were differences in arousal ratings observed in Probable PTSD participants as compared to the Trauma-exposed Group, irrespective of the type of stimuli, which suggests that contextual exposure impacts emotional arousal. This finding suggests that PTSD has a general effect on arousal for negative, neutral, and positive images. The Probable PTSD Group reported evidence of emotional processing dysfunction compared to the group of individuals who did not develop traumatic symptoms. Concerning the Probable PTSD participants there was substantial evidence that the effect of trauma enhanced emotional processing dysfunction. Hence, our findings provide insight into the challenges faced by this population and show that one chief factor causal to mental health complications in traumatized refugees and asylum seekers may be the alteration in emotion processing, such as: (a) enhanced emotional responses to earlier or current events (emotional reactivity), and (b) inadequate aptitude to cope with greater negative emotions and diminished positive emotions (emotion regulation). It is fundamental to understand how people with PTSD handle both negative and positive emotional information and how they regulate their own emotions [33,35,55,59]. Research has shown that PTSD in refugees and asylum seekers exposed to multiple traumas in their lifetime may promote different emotion and regulation processes compared to those who have been exposed to events of similar magnitude but who have not developed the disorder. Those who develop PTSD and evaluate positive images as less pleasant and negative images as more unpleasant, are on average increasingly activated and have less self-control in the face of visual stimuli of any value.

The emotional and affective states experienced by humans are multifaceted and shaded, and this feature makes them challenging to examine in a laboratory context to produce generalizable conclusions regarding the mechanisms underlying their source and maintenance. Even though these results can illuminate dysfunctions in PTSD attributable to weakened positive affect symptoms, undoubtedly supporting evidence for an impairment of the PVS as well as the NVS, more research is required to advance this field of study into one capable of producing clinically appropriate strategies that will expand trauma mental health care [14,28,29,32,57,69,71]. Therefore, our research also offers ideas for future studies demonstrating that interventional strategies should place an emphasis on desensitizing refugees with current or lifetime PTSD to traumatic memories, but specific attention should similarly be devoted to increasing their emotion regulation ability by managing symptoms of reduced positive affect/anhedonia.

## Figures and Tables

**Figure 1 behavsci-14-01195-f001:**
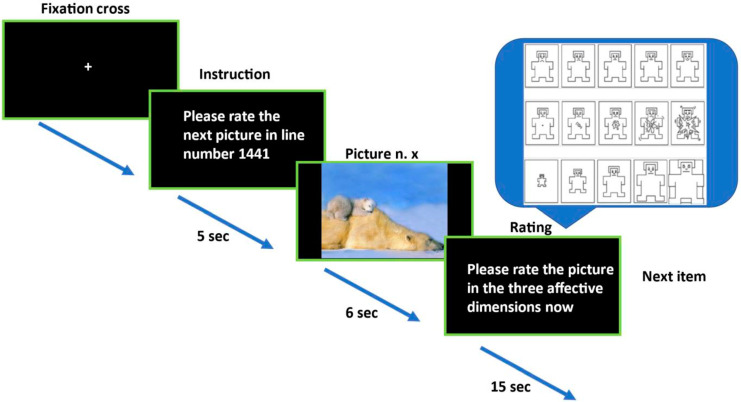
Study procedure. Each image was shown for six seconds after five seconds of organization. The participants were requested to complete the Self-Assessment Manikin (SAM) assessment within 15 s after seeing each photograph. SAM: (from top to bottom) the manikin symbols definite ratings of Valence (top), Arousal (mid), and Dominance (bottom). The scale ranges from 0 to 9 points.

**Figure 2 behavsci-14-01195-f002:**
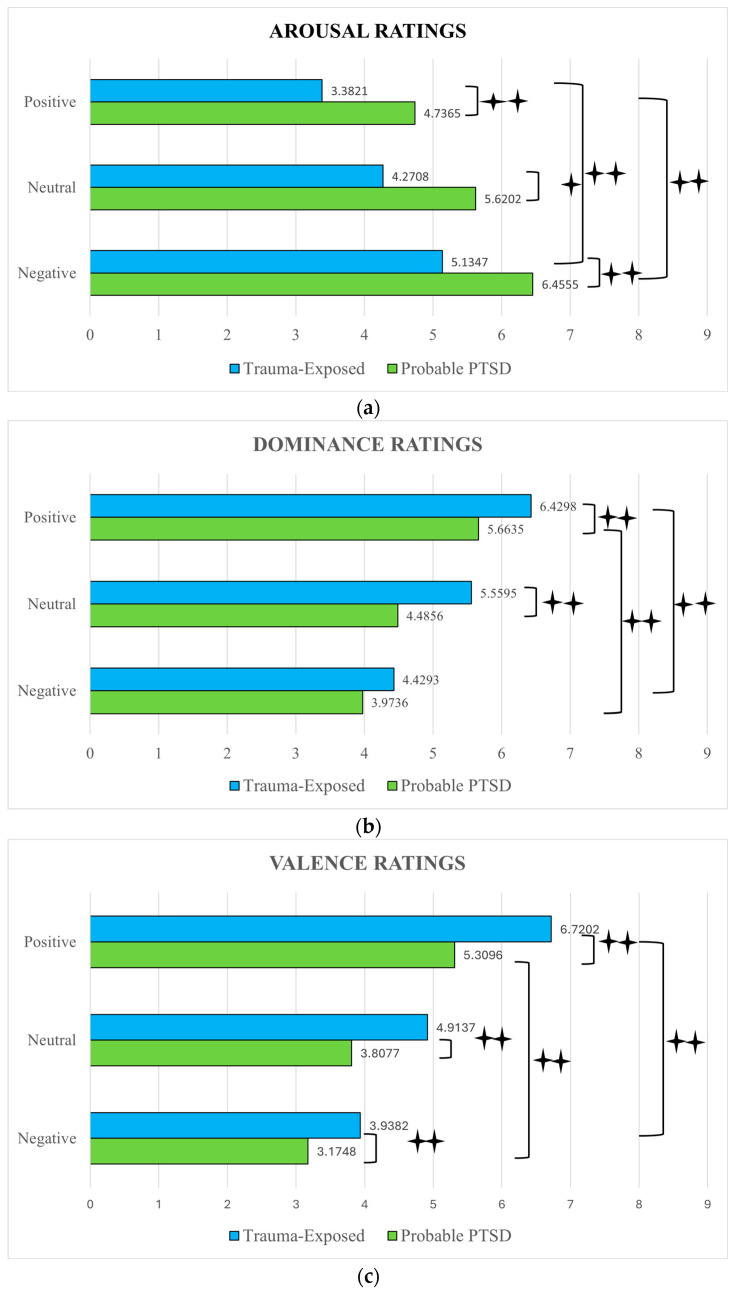
Self-Assessment Manikin arousal, dominance, and valence ratings of Probable PTSD and Trauma-exposed participants separated by image categories (positive, neutral, and negative), with arousal in the top panel (**a**), dominance in the middle panel (**b**), and valence in the bottom panel (**c**). ✦ indicate a statistical significance of ≤0.05. ✦✦ indicate a statistical significance of ≤0.01.

**Table 1 behavsci-14-01195-t001:** Scores from Part IV of the Harvard Trauma Questionnaire for measuring torture and trauma according to the DSM-5 criterion for PTSD (HTQ-5).

Main Variables	Probable PTSD (*n* = 42)	Trauma-Exposed (*n* = 26)	Group Comparison	
			*t* or *x*^2^	Sig. (Two-Tailed)
Age (years)	26.83 (7.66)	30.15 (7.90)	*t* = 0.171	0.090
Males *n* (%)	36 (85.7%)	22 (84.6%)	*x*^2^ = 0.015	0.901
Unmarried *n* (%)	31 (76.2%)	21 (80.7%)	*x*^2^ = 0.432	0.510
Muslim *n* (%)	29 (69.01%)	14 (53.8%)	*x*^2^ = 1.596	0.206
Middle school *n* (%)	18 (42.8%)	12 (46.1%)	*x*^2^ = 0.586	0.444
Asylum seekers/Refugee features				
Time since settlement in Italy (years)	3.33 (1.97)	4.15 (2.03)	*t* =1.64	0.100
Asylum seeker *n* (%)	33 (80.5%)	24 (92.3%)	*x*^2^ = 2.235	0.134
Potentially traumatic events experienced				
Travel towards Italy by boat *n* (%)	34 (80.9%)	20 (76.90%)	*x*^2^ = 0.159	0.689
Lack of water *n* (%)	14 (33%)	8 (30.7%)	*x*^2^ = 0.048	0.826
Lack of food *n* (%)	15 (35.7%)	8 (30.8%)	*x*^2^ = 0.175	0.675
Conflicts in country-of-origin *n* (%)	6 (14.6%)	6 (30.7%)	*x*^2^ = 0.854	0.355
Threats (e.g.: rape, torture) *n* (%)	28 (71.8%)	21 (84%)	*x*^2^ = 1.586	0.207
Torture or beate *n* (%)	25 (59%)	15 (57.7%)	*x*^2^ = 0.022	0.881
Sexual abuse *n* (%)	4 (10%)	1 (3%)	*x*^2^ = 0.759	0.383
Rape *n* (%)	15 (36.5%)	7 (26.9%)	*x*^2^ = 0.317	0.573
Serious injury *n* (%)	13 (31.7%)	9 (34.6%)	*x*^2^ = 0.294	0.587
Unnatural death of family/friend *n* (%)	24 (58.5%)	13 (50%)	*x*^2^ = 0.330	0.565
Witnessing family/friend murder *n* (%)	9 (21.4%)	8 (30.7%)	*x*^2^ = 1.651	0.198
Unnatural death of stranger *n* (%)	7 (17.1%)	7 (27%)	*x*^2^ = 1.033	0.309
Witnessing murder of stranger *n* (%)	10 (23.8%)	5 (19.2%)	*x*^2^ = 0.196	0.658
Time lapsed from exposure (years)	5.07 (3.37)	5.80 (3.84)	*x*^2^ = 0.82	<0.410
PTSD symptoms				
HTQ-5 scores (a)	2.43 (0.39)	1.46 (0.21)	*t* = 11.66	0.0001 **

(a) Scores from Part IV of the Harvard Trauma Questionnaire for measuring torture and trauma according to the DSM-5 criterion for PTSD (HTQ-5). Note that for English version the cut-off was 2.5 whereas for the Italian version was 2.00. ** indicate a statistical significance of ≤0.01.

**Table 2 behavsci-14-01195-t002:** ANOVAs results concerning group (Probable PTSD, Trauma-exposed) and pictures category (positive, neutral, negative).

Emotional Ratings	Effects	*F*-Statistics	*p*	η_p_^2^	Observed Power
Arousal					
	Main (Group)	*F*(1, 66) = 40.237	<0.001	0.379	1.000
	Main (Picture category)	*F*(1.782, 66) = 107.493	<0.001	0.620	1.000
	Group × Picture category	*F*(1.782, 66) = 0.012	0.982	0.000	0.052
Dominance					
	Main (Group)	*F*(1, 66) = 5.413	0.023	0.076	0.630
	Main (Picture category)	*F*(1.668, 66) = 79.812	<0.001	0.547	1.000
	Group x Picture category	*F*(1.668, 66) =2.231	0.121	0.033	0.406
Valence					
	Main (Group)	*F*(1, 66) = 30.388	<0.001	0.315	1.000
	Main (Picture category)	*F*(2, 66) = 245.717	<0.001	0.788	1.000
	Group × Picture category	*F*(2, 66) = 4.099	0.019	0.058	0.718

**Table 3 behavsci-14-01195-t003:** Correlations (Spearman’s correlation coefficients) between emotional ratings for pictures across all stimuli conditions, separately by group.

Probable PTSD		Arousal			Dominance			Valence		
		Negative	Neutral	Positive	Negative	Neutral	Positive	Negative	Neutral	Positive
**Arousal**	Negative	1	0.600 **	0.402 *	−0.495 *	−0.200	0.015	−0.709 **	−0.347	−0.335
	Neutral	0.600 **	1	0.425 *	−0.557 **	−0.672 **	−0.354	−0.274	−0.389 *	−0.276
	Positive	0.402 *	0.425 *	1	−0.322	−0.305	−0.304	−0.032	−0.026	−0.106
**Dominance**	Negative	−0.495 *	−0.557 **	−0.322	1	0.583 **	0.520 **	0.302	0.168	0.289
	Neutral	−0.200	−0.672 **	−0.305	0.583 **	1	0.556 **	0.004	−0.440 *	0.145
	Positive	0.015	−0.354	−0.304	0.520 **	0.556 **	1	−0.290	−0.030	0.100
**Valence**	Negative	−0.709 **	−0.274	−0.032	0.302	0.004	−0.290	1	0.394 *	0.401 *
	Neutral	−0.347	−0.389 *	−0.026	0.168	0.440 *	−0.030	0.394 *	1	0.501 **
	Positive	−0.335	−0.276	−0.106	0.289	0.145	0.100	0.401 *	0.501 **	1
**Trauma−Exposed**		**Arousal**			**Dominance**			**Valence**		
		**Negative**	**Neutral**	**Positive**	**Negative**	**Neutral**	**Positive**	**Negative**	**Neutral**	**Positive**
**Arousal**	Negative	1	0.613 **	0.335 *	−0.515 **	−0.292	−0.067	−0.368 *	0.084	0.074
	Neutral	0.613 **	1	0.668 **	−0.271	−0.335 *	−0.262	−0.207	−0.107	−0.190
	Positive	0.335 *	0.668 **	1	−0.115	−0.231	−0.282	0.041	0.006	−0.271
**Dominance**	Negative	−0.515 **	−0.271	−0.115	1	0.729 **	0.493 **	0.434 **	−0.018	0.058
	Neutral	−0.292	−0.335 *	−0.231	0.729 **	1	0.464 **	0.234	0.228	0.266
	Positive	−0.067	−0.262	−0.282	−0.493 **	0.464 **	1	0.138	−0.007	0.368 *
**Valence**	Negative	−0.368 *	−0.207	0.041	0.434 **	0.234	0.138	1	0.471 **	0.493 **
	Neutral	0.084	−0.107	0.006	−0.018	0.228	−0.007	0.471 **	1	0.609 **
	Positive	0.074	−0.190	−0.271	0.058	0.266	0.368 *	0.493 **	0.609 **	1

* Correlation significant on *p* < 0.05, ** Correlation significant on *p* < 0.01

## Data Availability

The dataset generated during and/or analyzed during the current study is available from the author on reasonable request.

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
