# Peer review of "Does Trauma Change the Way Individuals with Post-Traumatic Stress Disorder (PTSD) Deal with Positive Stimuli?"

_behavsci, 2024, doi:10.3390/bs14121195_

Round 1
Reviewer 1 Report
Comments and Suggestions for Authors
The article is well written, and the research was well designed, with congruent methodology, results, and discussion. It has social and academic relevance and adds new subsides to the field.
The abstract, however, needs to be improved. Suggestions:
Line 15 - The meaning of the acronym IAP has not been defined.
Lines 14-19 – This excerpt is statistically detailed. Still, it does not offer a greater understanding of the degree of contribution that the results of this research brought to academic knowledge on the subject. Replacing it briefly with what is much better written and detailed in lines 402-404, 442-445, 448-450, and 453-454 will attract greater attention from future readers/researchers, emphasizing the fair value of the academic effort made with this work.
The keywords "emotional numbing; Valence; arousal" are associated with the article's content inaccurately.
"Valence System, Emotional Regulation", for example, are suggestions to define the scope of this research better, increasing the chances of inclusion of this article in web search engines.
Reviewer 2 Report
Comments and Suggestions for Authors
I think it is a very interesting paper. It's true the sample is small and with a few number of women. I wonder if the differences between beliefs and cultures can influence the resilience of the Post-Traumatic Stress Disorder. The same between men and women. It could be also a future study in this line of research.
Interesting also that your study gives the possibility of diagnosing people who need help.
In the 2nd reference, it's DSM V. It's better DSM-5.
Thank you so much.
Best regards.
Reviewer 3 Report
Comments and Suggestions for Authors
Review report
The evaluated manuscript has a beginning that predicts an interesting, novel publication that will provide data that invites reading.
The introduction is presented consistently, laying the foundations for what is expected of an excellent work.
However, when you get to METHOD, the difficulties begin. The manuscript takes a confusing, uncertain path that leads to not easily and directly understanding what the authors intended.
It is not explained what type and design of research is carried out. This is very important, since readers will have a structure that will allow them to guide themselves correctly through the work. Therefore, it is suggested to include the type and design of the research.
A very important aspect is to declare and use the objective of the research as a guiding element. At least 4 objective statements have been identified. Sometimes one after the other (for example, reviewing lines 401-405). This takes away the strength of the work. The suggestion is that an objective statement be proposed, and that said statement be used verbatim every time the objective is mentioned. This gives coherence to the work. The following objective could be proposed: “to explore the emotional differences of valence (unpleasant/pleasant), arousal (calm/agitation) and dominance (domination/non-domination) in refugees diagnosed with PTSD from those who are not.”, as it aligns with what the authors are apparently looking for.
The study observes the use and reference to the DSM-V. Currently, there is a revised version of that same manual, the DSM-V-TR (APA, 2022). It is advisable to modify the citations/references and consult the revised text.
Another concern is regarding the instruments used. The manuscript does not evidence or present the psychometric properties of the IAPS for the preparation of norms in the selected population and the consequent validation of the instrument for its use in the research carried out. Nor does it do so with other instruments mentioned. The complete presentation of each mentioned instrument is suggested, including author or authors, year, objective of the instrument, qualification, classification, psychometric aspects, etc.
It is considered that the greatest difficulty is found in the presentation of the entire Method chapter. It is suggested, in addition to the specific aspects previously explained, that the organization of the chapter goes as follows:
- Research type and design
- Population, sample and sampling
- Instruments
- Procedure
- Data analysis
- Ethical aspects
Once the method chapter is reworked, the clarity, cohesion and methodological rigor of the manuscript can be significantly improved, bringing it closer to the international standards required for publication.
RESULTS
It is suggested that the presentation of tables or figures be carried out according to what was planned as a general objective and its specific objectives. Not being clear about these details constitutes a difficulty that prevents correctly evaluating the presentation of the results itself.
DISCUSSION
It must be reformulated according to what is proposed as a method and results.
REFERENCES
Although the manuscript presents 32.43% of references in indexed journals in the last 5 years, perhaps they make an effort to raise it to 40% or more. Furthermore, it is suggested to review the wording of each reference, to unify all of them into a single presentation style.
